# Microstructures and Mechanical Properties of Hybrid, Additively Manufactured Ti_6_Al_4_V after Thermomechanical Processing

**DOI:** 10.3390/ma14041039

**Published:** 2021-02-22

**Authors:** Susanne Hemes, Frank Meiners, Irina Sizova, Rebar Hama-Saleh, Daniel Röhrens, Andreas Weisheit, Constantin Leon Häfner, Markus Bambach

**Affiliations:** 1Access e.V., Intzestraße 5, 52072 Aachen, Germany; d.roehrens@access-technology.de; 2Otto Fuchs KG, Derschlager Straße 26, 58540 Meinerzhagen, Germany; frank.meiners@otto-fuchs.com; 3Brandenburg University of Technology, Konrad-Wachsmann-Allee 17, 03046 Cottbus, Germany; sizova@b-tu.de; 4Fraunhofer Institute for Laser Technology (ILT), Steinbachstraße 15, 52074 Aachen, Germany; rebar.hama-saleh@ilt.fraunhofer.de (R.H.-S.); andreas.weisheit@ilt.fraunhofer.de (A.W.); constantin.haefner@ilt.fraunhofer.de (C.L.H.); 5ETH Zürich, Technoparkstrasse 1, 8005 Zürich, Switzerland; mbambach@ethz.ch

**Keywords:** laser directed energy deposition (L-DED), thermomechanical processing (TMP), Ti_6_Al_4_V, hybrid manufacturing

## Abstract

In the present study, we propose a hybrid manufacturing route to produce high-quality Ti_6_Al_4_V parts, combining additive powder laser directed energy deposition (L-DED) for manufacturing of preforms, with subsequent hot forging as a thermomechanical processing (TMP) step. After L-DED, the material was hot formed at two different temperatures (930 °C and 1070 °C) and subsequently heat-treated for stress relief annealing. Tensile tests were performed on small sub-samples, taking into account different sample orientations with respect to the L-DED build direction and resulting in very good tensile strengths and ductility properties, similar or superior to the forged material. The resulting microstructure consists of very fine grained, partially globularized alpha grains, with a mean diameter ~0.8–2.3 µm, within a beta phase matrix, constituting between 2 and 9% of the sample. After forging in the sub-beta transus temperature range, the typical L-DED microstructure was no longer discernible and the anisotropy in tensile properties, common in additive manufacturing (AM), was significantly reduced. However, forging in the super-beta transus temperature range resulted in remaining anisotropies in the mechanical properties as well as an inferior tensile strength and ductility of the material. It was shown, that by combining L-DED with thermomechanical processing in the sub-beta transus temperature range of Ti_6_Al_4_V, a suitable microstructure and desirable mechanical properties for many applications can be obtained, with the advantage of reducing the material waste.

## 1. Introduction

Ti_6_Al_4_V is a lightweight, high-strength, two-phase (alpha-beta) titanium alloy, and due to its favorable mechanical properties, it is used in many industries (e.g., aviation and medical industries) as a high performance material [1]. Among its most valuable properties are, besides its high strength and low density, a good ductility as well as corrosion resistance and an excellent performance at elevated temperatures (up to 550 °C) [2]. However, most of these properties are highly dependent on the microstructure of the material, which itself is determined by the thermal history and in particular the cooling rate from the beta phase field [3,4]. Therefore, the final microstructure, grain-size and mechanical properties are highly sensitive to the manufacturing route and the applied post-processing steps [5,6,7]. Traditionally, the material is produced by hot forging with subsequent extensive post-processing (e.g., machining) needed to achieve the desired part dimensions and properties, leading to a material yield in the range of only ~3–20% [8,9,10,11], and accounting for the majority of the manufacturing costs. However, new production chains—such as additive manufacturing (AM)—are emerging and enabling the near net shape (NNS) production of high-performance parts, with a material yield close to 100% [12]. On the other hand, in AM-production, depending on the process-parameters, the build-up speed is still low and for series production as well as for the production of larger parts with simple geometries, conventional forging production chains are still favorable over AM [13]. The main advantage of AM in the production cycle lies in its flexibility to manufacture complex shapes and structures, with a significantly reduced material waste. Key industries, where AM delivers significant advantages over conventional production, are, therefore, repair industries or industries where customized design objects and mass customization—such as dental supply—are needed, and where conventional production is very costly.

In general, following AM, several post-processing steps—such as hot isostatic pressing (HIP) to close internal porosities, machining for surface roughness removal or surface modifications via electrical discharge coating (EDC) or shot-peening, to improve mechanical properties and in particular, increase the hardness, wear resistance and corrosion resistance of the surface layer [14], as well as heat treatments to reduce residual stresses, resulting from the fast cooling during the AM-process—are still necessary to achieve the desired mechanical properties for many applications [13,14,15,16]. The surface roughness of additively manufactured (L-DED) parts clearly depends on the laser spot diameter, the powder particle size distribution (in the case of powder involved), as well as the inclination of the product towards the laser beam or energy source [17]. Measured surface roughness (Ra) values for AM parts range, depending on the process, from 0.24 to 131 µm [12,17]. For parts produced in the present study, after L-DED, a surface roughness (Ra) between 15 and 20 µm was measured. Combining suitable post-processing steps, the tensile strength and ductility of forged materials can even be exceeded by AM products, but with anisotropy in both tensile strength and ductility remaining a major problem, despite the application of post-AM sub- or super-beta transus heat treatments [11]. Alternatively, hot forging may be used as a thermomechanical processing treatment, thus combining the advantages of forming with AM techniques in a hybrid process chain. Recently, it was shown that for the Ti_6_Al_4_V alloy, produced by laser powder bed fusion (LPBF) or L-DED, combining AM with forging, the hot deformation behavior of the material at temperatures between 850 and 1000 °C was improved, and a finer microstructure as well as lower flow stresses, compared to conventionally processed materials, were obtained [18,19,20]. While several studies [21,22] confirm the advantages of combining AM with hot working, there exist no systematic studies on the mechanical performance of hot forged, additively manufactured Ti_6_Al_4_V. The present study investigates the use of L-DED—providing preforms —combined with hot forging as a thermomechanical processing (TMP) step, to produce parts with favorable mechanical properties in a hybrid process chain and at the same time, reduce the material waste. AM preforms were hot deformed at different conditions and subjected to different heat treatments. Finally, extensive investigations on the obtained microstructures and static mechanical properties were performed, showing satisfactory results, concerning both strength and ductility as well as isotropy of the obtained tensile properties. The nomenclature of the abbreviations used in the present study are given in Table 1.

## 2. Materials and Methods

A total of three different processing routes were examined as thermomechanical treatments after L-DED for the hybrid production chain: (1) alpha-beta forging, followed by beta annealing, (2) alpha-beta forging, followed by stress relief annealing and (3) beta forging, followed by stress relief annealing. In all cases, the hybrid samples consisted of a forged base plate with dimensions of 100 × 100 mm and a thickness of 10 mm, to which 50 × 50 × 100 mm^3^ of Ti_6_Al_4_V material was added by L-DED (Figure 1c).

During the L-DED process, the powder material was fed with argon gas through a coaxial powder nozzle (Coax50; Fraunhofer-Institut für Lasertechnik; Aachen, Germany; Figure 1a), providing an extended stand-off (i.e., distance between nozzle and substrate), into the melt pool, created by the laser beam. The laser beam source (LDF 12000-100; Laserline GmbH; Mülheim-Kärlich, Germany) is a fiber-coupled diode laser with a maximum laser power of 12 kW. The diode laser head consists of eight diode stacks in four wavelengths (910, 940, 980 and 1030 nm). Using polarization and wavelength coupling, the laser radiation is coaxially superimposed and coupled into an optical fiber (HighYag Lasertechnologie GmbH; Kleinmachnow, Germany) with a core diameter of 1000 μm and a numerical aperture of 0.22. This laser beam source has a beam quality of 100 mm mrad. After coupling, the laser beam hits the collimating lens and is rectified to the zoom optic (KUKA Industries GmbH and Co. KG; Würselen, Germany). A Sulzer Metco Single 10-C (Sulzer Metco Coatings GmbH; Salzgitter, Germany) powder feeder was used, with argon as a carrier gas, to regulate the powder feed rate. To achieve high powder feed rates, a high deposition-rate nozzle designed by Fraunhofer ILT was used for performing the experiments. The inert gas stream inside the nozzle was used for additional protection of the melt pool from oxidation and the content of oxygen in the surrounding atmosphere was measured below 180 ppm during the whole process. The scanning speed of the substrate can be adjusted via CNC (computer numerical control). Within the scope of process development, experiments on L-DED were carried out using a 3-axis handling system (Schuler Held GmbH; Heusenstamm, Germany). The tool path for the build-up of volumes was generated by LMDCAM2^®^ (v1.1.6) with a bi-directional scanning strategy and the orientation of the path was alternating by a 90° angle during the L-DED build-up, as illustrated in Figure 1b. Table 2 summarizes the L-DED build parameters, which were determined in a process study, to optimize the resulting density of built volumes, their geometry, oxygen content and the time needed for build-up [23].

In the following, the L-DED built samples were forged using a hydraulic Eumuco 10 MN/1 kilo-ton press (SMS Eumuco GmbH, Elkenroth, Germany), with mold temperatures (upper and lower mold) ~440 °C. A punch speed of 30 mm/s was applied and an oil-graphite mixture was used as a lubricant. Samples were deformed up to 25–30% as well as 50% compression, respectively, at two different deformation temperatures, one in the alpha-beta phase field (~930 °C) and one in the beta phase temperature regime (~1070 °C) of Ti_6_Al_4_V. Afterwards, either beta annealing at 1050 °C for 3 h, followed by stress relief annealing at 710 °C for 6 h and subsequent cooling in air, or stress relief annealing at 710 °C for 6 h, followed by cooling in air, was used as a heat treatment. Figure 2 illustrates the produced samples after the hot deformation process and Table 3 summarizes the TMP routes, applied in the present study for hot deformation and heat treatment.

Following the thermomechanical treatments, sample macrostructures were examined using a light microscope. Afterwards, samples were cut into smaller sub-samples for mechanical analyses (i.e., tensile tests), oriented 0° (vertical), 45° and 90° (horizontally), with respect to the L-DED build direction, as illustrated in Figure 3, where locations of tensile specimens, as well as an example of a tensile sample after failure, are given. Tensile tests were carried out at room temperature, in accordance with DIN-EN-ISO 6892-1 [24], on an electro-mechanical tensile testing machine. After failure, small specimens were cut from the horizontal sub-samples for microstructural analyses, with the direction of view perpendicular to the L-DED build direction. Microstructural analyses were carried out on two Zeiss field emission scanning electron microscopes (Zeiss Gemini FE-SEM ULTRA55 and 1540XB; Carl Zeiss Microscopy GmbH, Oberkochen, Germany), equipped with back-scattered electron (BSE) detector (Oxford TETRA; Oxford Instruments GmbH, Wiesbaden, Germany), energy dispersive X-ray (EDX) detector (Oxford ULTIMMAX170; Oxford Instruments Analytical GmbH, Abingdon, England) and an Oxford EBSD system, with Nordlys EBSD camera (Oxford Instruments plc., Abingdon, UK) For BSE and EDX analyses, an electron acceleration voltage of 15 kV was used, with a working distance of about 10.5 mm. The energy range used for EDX analysis was 20 keV. The amount of beta phase fraction was measured via image analysis techniques, using ImageJ on BSE-images and applying thresholding techniques on the different grey-scale values of the alpha and beta phases in the respective images. Moreover, these results were compared to and confirmed by EBSD analyses.

Moreover, instrumented micro-indentation was carried out on the horizontal sub-samples, using an Anton Paar AP-MHT3 micro-indenter (Anton Paar Germany GmbH, Ostfildern, Germany). The maximum load was controlled between 0.5 and 2 Newtons and the loading and unloading rates were between 1 and 2 N/min. For the calculation of mean instrumented hardness, matrices of 25–30 indents were placed on the sample surfaces with distances of about 100 µm between each indent in x- and y-direction. The loading direction during micro-indentation experiments was perpendicular to the initial L-DED build direction and was, therefore, also perpendicular to the direction of compressive stress during the hot forming of bulk samples. Finally, the modulus of elasticity was calculated from the unloading curves using the Oliver–Pharr method, as described in [25].

## 3. Results and Discussion

### 3.1. Macrostructures

Macrostructures of hot deformed samples are summarized in Figure 4, showing significant differences between the different processing routes.

In Figure 4a, prior to hot deformation (i.e., as built), elongated primary beta grains with their long axes oriented parallel to the L-DED build direction (vertical in the images below)—in accordance with the largest temperature gradient during L-DED manufacturing and cooling from the beta phase field—are visible, as well as a heterogeneous macrostructure, changing from smaller beta grains at the bottom (<1 mm) to very large, elongated beta grains at the center towards the top part of the sample (up to several mm length), which is the result of the heating-up of L-DED samples during the manufacturing of Ti_6_Al_4_V, due to the rather low heat conductivity of the material [26,27]. In Figure 4b, after alpha-beta forging (~930 °C) to a compression of 50%, followed by beta annealing (~1050 °C), equiaxed prior beta grains with a mean diameter ~3.3 mm are visible, as is a rather homogeneous macrostructure. In Figure 4c, the sample deformed in the beta phase regime (~1070 °C), to a compression of 50%, followed by stress relief annealing, shows a heterogeneous macrostructure, which appears highly distorted, with prior beta grain boundaries still discernible. However, the deformation of prior beta grains, i.e., a compression in the direction parallel to the initial L-DED build direction, is also clearly noticeable. Furthermore, deformation structures and an elongation of the prior beta grains in the direction perpendicular to the direction of compressive stress, are visible. Moreover, in the uppermost central part of this sample and at its bottom (central part), the primary L-DED macrostructure still seems to be maintained.

In Figure 4d,e, for samples forged in the alpha-beta phase regime, to a compression of 25–30% (Figure 4d), and to 50% compression (Figure 4e), followed by stress relief annealing, the primary L-DED macrostructure is still discernible; however, it is only discernable in the outer regions of the samples (top, bottom and sides), whereas at the samples’ centers, a completely different macrostructure is visible, which appears much finer. Such effects may be ascribed to the inhomogeneous deformation behavior, which can often be found in hot compression tests, due to friction forces at the interfaces between sample and molds, as well as barreling effects [28].

### 3.2. Microstructural Analysis

For microstructural analyses, BSE-SEM images were taken at magnifications of 500× (scale bar = 50 µm) and 5000× (scale bar = 5 µm), both from the centers as well as from the side regions of L-DED manufactured samples, after hot deformation and heat treatments (Figure 5e–r). For reference, in Figure 5a–d, microstructures of an as-built sample, illustrating the differences between the sample center (Figure 5a,b) and the side regions (Figure 5c,d), are shown. Whereas alpha lamellae at the side regions are considerably thicker (~1.6 µm on average) and also an increased amount of beta phase fraction (~5%) was measured, at the sample center the thickness of alpha lamellae is only about 0.8 µm on average and the measured beta phase fraction is ~2%. Furthermore, at the sides (Figure 5c), the primary beta grain-size is significantly larger (up to 500 µm in diameter), compared to the center of the sample (Figure 5a), showing a primary beta grain width ~100 µm (average). These differences are interpreted to be due to the very heterogeneous heat distribution and cooling velocities from the beta phase field during L-DED manufacturing, resulting in very heterogeneous thermal histories as well as micro- and macrostructures of the material [29,30] (see also Figure 4a). Samples deformed in the alpha-beta phase regime (~930 °C) with subsequent beta heat treatment at 1050 °C (Figure 5e–h) show a homogeneous Widmanstätten (alpha-beta) microstructure at the sample center and at its sides, with alpha lath thicknesses ~0.9 µm (average) and a measured beta phase fraction ~2.5%. This microstructure is the result of a complete recrystallization of the material in the beta phase field during heat treatment. However, a slight coarsening of the alpha laths presumably took place again during the final heat treatment at a temperature of 710 °C [31]. Moreover, this final heat treatment probably resulted in a slight increase in the beta-phase fraction in between the alpha laths (Figure 5e–h). However, sample no. 4, forged in the beta phase regime (~1070 °C), also underwent recrystallization with simultaneous deformation to ~50% compression. The measured alpha lath thickness is ~1.1 µm in the center of this sample (Figure 5i,j) and ~0.8 µm near the lateral surface of the sample (Figure 5k,l). The coarsening of the alpha laths at the sample center is interpreted to be due to heat dissipation, leading to an increase in the alpha laths’ thickness as a function of the temperature increase. Another factor could be due to the sample compression [32]. The measured remaining beta phase fraction is ~2% at this samples’ side regions (Figure 5k,l) and ~3% at the sample center (Figure 5i,j). Samples deformed in the alpha-beta temperature range, followed by stress relief annealing heat treatment at 710 °C (Figure 5m–r) show partially globularized microstructures, with beta phase fractions ~9% and a mean (globular) alpha grain-size ~2.3 µm for the sample compressed to about 50% (Figure 5m–p). The microstructure appears homogeneous between the sample center (Figure 5m,n) and the regions close to the lateral surface (Figure 5o,p), although some differences were observed between the sample center and side regions on the macroscale (Figure 4d,e).

The thermomechanical treatment within the alpha-beta (or sub-beta transus) temperature field resulted in: (i) coarsening of the Widmanstätten alpha laths, (ii) a transition towards a lamellar microstructure, (iii) an increase in the beta phase fraction, as well as (iv), the on-set of globularization processes, accompanied by lamellar kinking (Figure 5r) and continuous dynamic recrystallization of the alpha-phase [6,33]. Another sample, which underwent this thermomechanical treatment, was compressed to ~25–30% only, showing less globularized alpha microstructures and still a more lamellar appearance (Figure 5q,r), with an alpha lamellae thickness ~2.1 µm and a beta phase fraction ~9%.

### 3.3. Mechanical Testing

#### 3.3.1. Tensile Tests

Tensile test results are summarized in Table 4, showing mean ultimate tensile strengths (UTS; R_m_) and the amount of stress resulting in a plastic strain of 0.2% (R_p0.2_). Moreover, values for the total strain (plastic plus elastic) of the gauge length at the moment of failure, as a percentage of the original gauge length before deformation (A5), and the percentage of reduction of the cross-sectional area of the gauge (Z), are given.

The results show the highest and most isotropic strength and ductility for samples forged in the alpha-beta regime to a compression of 50%, followed by stress relief annealing (sample no. 2). Although samples forged in the alpha-beta range to a compression of 50%, followed by beta annealing (sample no. 1), show a higher overall ultimate tensile strength (R_m_) of up to 938 MPa and R_p0.2_ values up to 864 MPa, these properties strongly depend on the sample orientation, showing lower values in the 45° direction (R_m_ = 899 MPa and R_p0.2_ = 831 MPa). Furthermore, these higher strengths come at the expense of ductility, with an elongation at break of only 4.5–6.1% (A5), which is significantly lower compared to the rest of the samples analyzed. The highest ductility, i.e., an elongation at break of about 14.5%, was measured for samples forged in the alpha-beta regime, followed by stress relief annealing, in the direction parallel to the initial L-DED build-up and parallel to the direction of compression, whereby both the ductility and strength seem to increase with the degree of compression (Table 4). Samples forged in the beta phase field (sample no. 4), followed by stress relief annealing, show the lowest strengths, and with a strong anisotropy, whereby the lowest values of R_p0.2_ (~749 MPa) and R_m_ (~833 MPa) were measured for samples oriented parallel to the l-DED build direction and the highest for samples oriented horizontally (R_m_ = 884 MPa and R_p0.2_ = 795 MPa). The elongation at break and thus the ductility of these samples also shows a strong anisotropy, with maximum values ~7.9% for the horizontally oriented specimens and values of only ~4.3% for the 45° oriented samples. These results indicate that different processing routes result in different directional dependencies of mechanical properties. In principle, any anisotropies resulting from the L-DED process should have been eliminated, due to the thermomechanical processing and heat treatments applied in the present study. However, the equiaxed beta grains of sample no. 1 seem to have inherited some of the internal texture of the prior columnar beta grains during grain growth and recrystallization. A phenomenon that has already been postulated by [34], suggests that a “relatively weaker anisotropy is inherited as a result of sub-grain growth mechanisms during the transforming from columnar to equiaxed beta grains”. In general, after both beta annealing and beta forging (samples S1 and S4), anisotropies in the tensile properties still seem to be maintained, which might be the result of some inherited texture in the beta grains, resulting from the L-DED manufacturing [34,35]. The as-built material (Figure 6) exhibits a strong tensile anisotropy due to the columnar grain morphology and the strong texture of <001>_β_ parallel to the build direction [23], resulting in an orientation of many of the preferential slip systems (i.e., <111>_β_ on {110}_β_ in bcc-beta, but due to Burger’s orientation relationship (BOR) [36], also <11–20>_α_ on {0001}_α_ in hcp-alpha, i.e., basal slip) in alpha and beta Ti_6_Al_4_V at 45° towards the direction of build-up [34]. Therefore, often lower tensile ductility is found at a 45° orientation towards the build direction. However, this aspect needs further investigation in the future.

In contrast, after alpha-beta forging, followed by stress relief annealing (samples S2 and S3), both tensile strength and ductility seem to be nearly isotropic, suggesting that this processing route is favorable for structural components where isotropic mechanical properties are needed, as is the case for many aerospace and other engineering applications. Further investigations on the material’s dynamic mechanical properties (e.g., fatigue endurance limit) are, however, necessary to complete the picture.

Comparing the tensile test results of the present study to tensile properties obtained by the authors in a previous study [23] (Figure 6) on L-DED manufactured samples, consisting of a forged base plate to which material was added by L-DED and subsequently heat-treated (i.e., stress relief annealed or beta annealed), shows that in particular the processing routes combining sub-beta transus forming/forging with stress relief annealing result in a significantly improved isotropy of tensile properties (strength and plasticity). Comparing the results of the present study to norm-values for alpha-beta forged and stress relief annealed material (ASTM B381: R_m_ = 895 MPa, R_p0.2_ = 828 MPa, A5 = 10% and Z = 25%), shows that in particular the processing route no. 2 exceeds these norm values for all instances given. However, norm-values for beta-forged and beta-annealed material (ASTM B367: R_m_ = 895 MPa, R_p0.2_ = 825 MPa and A5 = 6%) were not reached in the present study (Table 4). Sample no. 4 (beta forging) does not fulfil any of the criteria given, and although sample no. 1 (beta annealing) does exceed the norm-values for R_m_ and R_p0.2_, an elongation at break (A5 value) of 6% was only reached for the vertical tensile samples, oriented parallel to the build direction, which could again be due to the texture inheritance, as described above [34].

#### 3.3.2. Micro-Indentation

Micro-indentation test results (Table 5) show the highest instrumented hardness for sample no. 1, followed by samples no. 2/4 and 3, which is in good agreement with the results obtained by tensile testing, showing the highest strength and lowest plasticity (R_p0.2_) as well as elasticity for sample no. 1 (followed by sample set no. 2). Differences in the measured instrumented elastic moduli fall within the range of standard deviation (~3–6 GPa) for these measurements. Since the hardness in additively manufactured Ti_6_Al_4_V is mainly a function of the martensite content [12] as well as the oxygen content, which was kept below 180 ppm during the whole AM process in the present study, due to the applied heat treatments and thermomechanical processing above the temperature range of martensite decomposition (~600–900 °C) [32], no significant increase in the measured hardness of the material was observed. However, material hardness in Ti64 is also a function of the alpha lamellae thickness and the β-phase fraction, showing an inverse correlation with both. Therefore, the lower hardness measured for samples no. 2 and 3 seems reasonable with regard to the microstructural observations made in the present study (Figure 5m–r).

## 4. Conclusions

In the present study, a hybrid process chain is presented, combining additive manufacturing (L-DED) with hot forging of Ti_6_Al_4_V, thus reducing the material waste during manufacturing, compared to the conventional production (i.e., forging) of Ti_6_Al_4_V components.

(1) Via thermomechanical processing (i.e., forming) in the sub-beta transus temperature range (~930 °C) of the alloy, followed by stress relief annealing heat treatment, improved tensile strength and ductility, compared to L-DED as-built and stress relief annealed materials were obtained, particularly with regard to the anisotropy of tensile strength and plasticity. Isotropy of tensile properties is of importance for many applications of Ti_6_Al_4_V, for example as structural components in the aerospace industry.

(2) However, thermomechanical processing in the super-beta transus temperature range, following L-DED, does not appear suitable for obtaining the desired mechanical properties of the hybrid manufactured Ti_6_Al_4_V samples, which is postulated to be due to several factors, such as excessive beta grain growth above the beta transus temperature [6,11], as well as inherited texture within equiaxed beta grains, resulting from the columnar and textured structure in as-built samples [34].

(3) Since this is a preliminary study for industrial application of the presented hybrid manufacturing route, further studies on the dynamic mechanical properties (e.g., fatigue endurance limit) as well as research on the reproducibility of the results obtained in the present study are necessary to support the results presented here.

(4) Another question, anticipated for further studies, is whether the total manufacturing costs are reduced by combining AM with heat forming in a hybrid production chain. Further research will be conducted on demonstrator parts for the aerospace industry.

## Figures and Tables

**Figure 1 materials-14-01039-f001:**
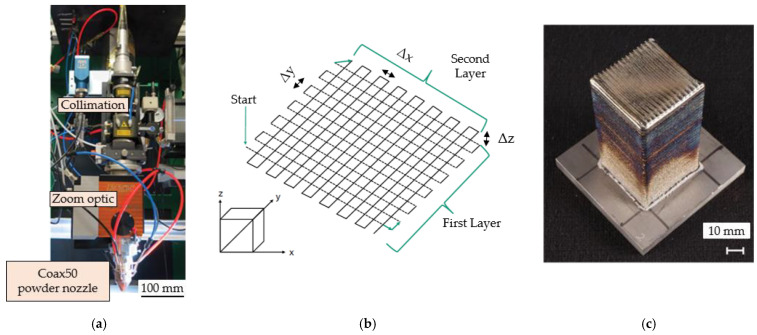
(**a**) L-DED set-up used in the present study, (**b**) illustration of the applied scanning strategy (schematic sketch) used for L-DED manufacturing, and (**c**) L-DED as-built sample, prior to hot deformation.

**Figure 2 materials-14-01039-f002:**
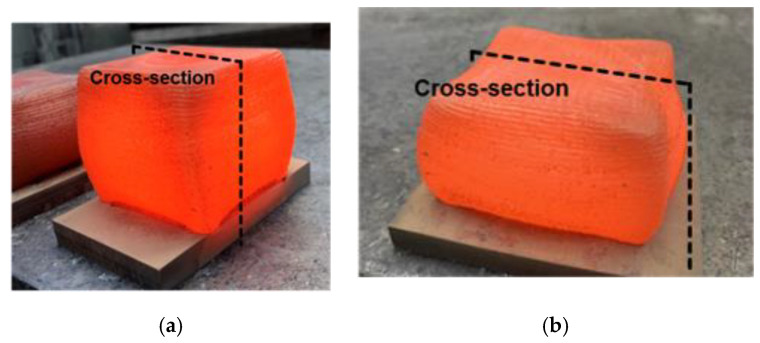
(**a**) L-DED built sample after hot forging at a temperature of 930 °C, deformed to about 25% compression (cross-section Figure 4d indicated), (**b**) L-DED built sample after hot forging at ~930 °C, deformed to about 50% compression (cross-section Figure 4e indicated).

**Figure 3 materials-14-01039-f003:**
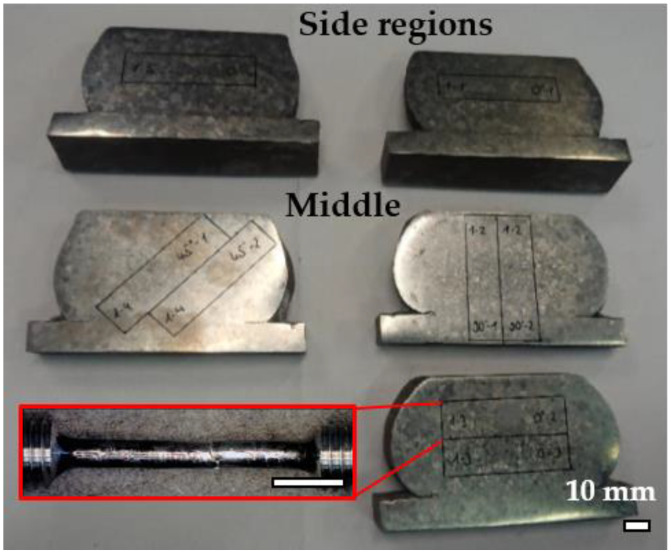
Illustration of locations, orientations and dimensions of tensile specimens analyzed in the present study (scale bars are 10 mm); example demonstrated for sample no. 1.

**Figure 4 materials-14-01039-f004:**
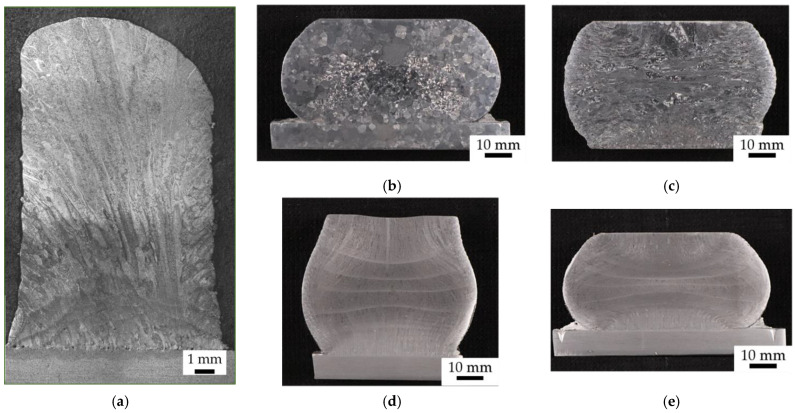
(**a**) L-DED as-built sample, prior to hot deformation; (**b**–**e**) macrostructures of forged samples: (**b**) after alpha-beta forging to a compression of 50%, followed by beta annealing; (**c**) after beta forging to 50% compression, followed by stress relief annealing, (**d**) after alpha-beta forging to 25–30% compression, followed by stress relief annealing and (**e**) after alpha-beta forging to a compression of 50%, followed by stress relief annealing.

**Figure 5 materials-14-01039-f005:**
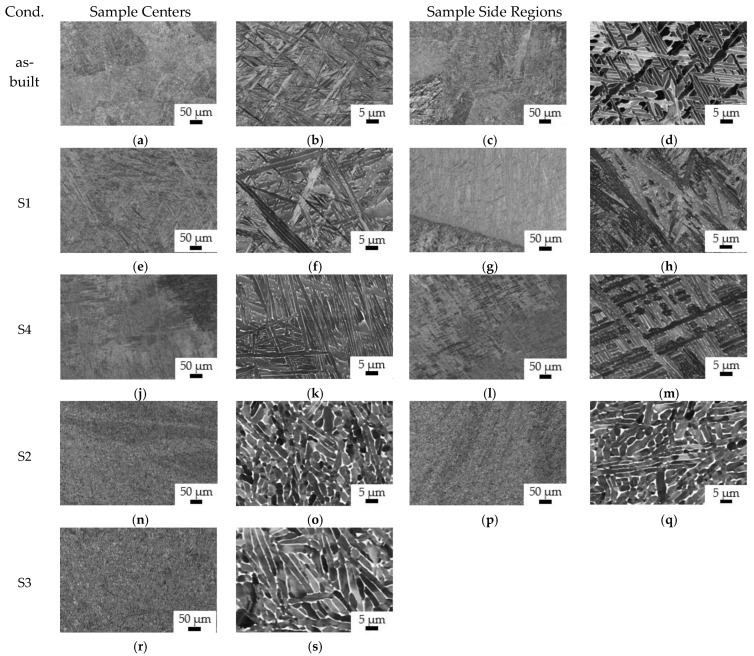
Microstructures of hot deformed samples, as visualized by BSE-SEM at magnifications of 500× and 5000×; scale-bars are 50 µm and 5 µm, respectively. (**a**–**d**) For reference, micrographs of an un-deformed, as-built L-DED sample are shown. (**e**–**h**) Microstructures of alpha-beta forged and beta heat treated material (S1). (**i**–**l**) Microstructures of beta forged and stress relief annealed sample (S4). (**m**–**p**) Microstructures after alpha-beta forging to 50% compression (S2), and to 25–30% compression (S3) (**q**–**r**), respectively, both followed by stress relief annealing heat treatment.

**Figure 6 materials-14-01039-f006:**
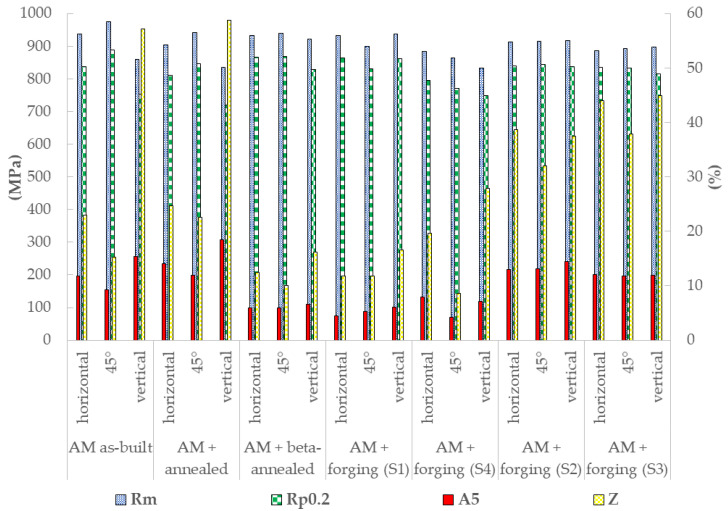
Comparison of tensile properties of hybrid manufactured (AM + forging) samples with L-DED manufactured Ti_64_ tensile properties without thermomechanical processing (TMP) (data from [23]).

**Table 1 materials-14-01039-t001:** Nomenclature used in the present study.

Abbreviation	Explanation
**L-DED**	laser directed energy deposition
**TMP**	thermomechanical processing
**AM**	additive manufacturing
**NNS**	near net shape
**HIP**	hot isostatic pressing
**EDC**	electrical discharge coating
**Ra**	arithmetic average deviation of the measured surface roughness profile from the center line of the measured profile
**LPBF**	laser powder bed fusion
**CNC**	computer numerical control
**BSE**	back scattered electron
**EDX**	energy dispersive X-ray
**UTS (R_m_)**	ultimate tensile strength or yield strength
**R_p0.2_**	stress resulting in a plastic strain of 0.2%
**A5**	percentage of plastic plus elastic strain of the gauge (length) at the moment of failure relative to the original gauge length
**Z**	percentage of reduction of the cross-sectional area of the gauge
**BOR**	Burger’s orientation relationship

**Table 2 materials-14-01039-t002:** L-DED parameters, used in the present study for the manufacturing of preforms.

Laser Spot Diameter (mm)	Laser Power (W)	Scanning Speed (mm/min)	Powder Mass Flow (g/min)	Shielding Gas Flow (L/min) (Argon)	Carrier Gas Flow (L/min) (Argon)	Δx/Δy (mm)	Δz (mm)	Stand-off (mm)
3.0	1680	1500	8.5	10	7	1.5	0.85	16

**Table 3 materials-14-01039-t003:** Applied thermomechanical treatments after L-DED manufacturing.

Sample Number	Forging	Heat Treatment
**1**	Alpha-beta Forging at 930 °C up to 50% compression	Beta annealing at 1050 °C for 3 h + stress relief annealing at 710 °C for 6 h + cooling in air
**2**	Alpha-beta forging at 930 °C up to 50% compression	Stress relief annealing at 710 °C for 6 h + cooling in air
**3**	Alpha-beta forging at 930 °C up to 25–30% compression	Stress relief annealing at 710 °C for 6 h + cooling in air
**4**	Beta forging at 1070 °C up to 50% compression	Stress relief annealing at 710 °C for 6 h + cooling in air

**Table 4 materials-14-01039-t004:** Tensile test results. For thermomechanical treatments of samples analyzed, see Table 3.

Sample No.	Orientation	R_m_ (Mpa)	R_p0.2_ (Mpa)	A5 (%)	Z (%)	No. Samples Analyzed
**1**	Horizontal	933	864	4.5	11.8	4
	45°	899	831	5.3	11.8	2
	Vertical	938	861	6.1	16.6	2
**2**	Horizontal	912	840	13.1	38.6	4
	45°	916	844	13.1	32.1	2
	vertical	916	838	14.5	37.5	2
**3**	Horizontal	886	836	12.1	43.9	3
	45°	893	833	11.8	37.9	2
	Vertical	897	816	11.9	44.9	2
**4**	Horizontal	884	795	7.9	19.7	4
	45°	864	771	4.3	8.6	2
	Vertical	833	749	7.2	27.9	2

**Table 5 materials-14-01039-t005:** Micro-indentation test results.

Sample Reference	Mean Instrumented Hardness (GPa)	Mean Instrumented Elastic Modulus (GPa)
1	3.9 (± 0.2)	132 (± 6)
2	3.7 (± 0.4)	126 (± 5)
3	3.6 (± 0.1)	130 (± 3)
4	3.7 (± 0.1)	128 (± 6)

## Data Availability

Data is contained within the article.

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
