# Peer review of "Microstructures and Mechanical Properties of Hybrid, Additively Manufactured Ti6Al4V after Thermomechanical Processing"

_materials, 2021, doi:10.3390/ma14041039_

Round 1

Reviewer 1 Report

The present investigation is an interesting work. However, there are many points addressed here should be revised very carefully. (1) The English writing is poor. I suggest that the authors revise the spelling and grammar in the whole text. (2) The key words should not be so many, the abstract and conclusion need to be improved. (3) 3.2.Microstructural analysis. "Whereas alpha lamellae at the side regions are considerably thicker (~1.6µm on average) and also an increased amount of beta phase fraction (~5%) was measured." How is the increased amount of beta phase fraction calculated? (4) 3.3.Mechanical testing. The tensile test results and micro-indentation test results should be analyzed, and the test results cannot be simply listed.

Author Response

Answers to Reviewer no. 1:

Reviewer

The present investigation is an interesting work. However, there are many points addressed here should be revised very carefully.

(1) The English writing is poor. I suggest that the authors revise the spelling and grammar in the whole text.

Response 1: The spelling and grammar of the whole article were revised.

(2) The key words should not be so many, the abstract and conclusion need to be improved.

Response 2: The key words have been reduced and the abstract and conclusions were improved.

(3) 3.2.Microstructural analysis. "Whereas alpha lamellae at the side regions are considerably thicker (~1.6µm on average) and also an increased amount of beta phase fraction (~5%) was measured." How is the increased amount of beta phase fraction calculated?

Response 3: The amount of beta phase fraction was measured via image analysis techniques, using ImageJ on BSE-images and applying simple thresholding techniques (in ImageJ) on the different grey-scale values of the alpha and beta phases in the respective BSE-images. Moreover, these results were compared to and confirmed by EBSD analyses. I added these information to the methods section.

(4) 3.3.Mechanical testing. The tensile test results and micro-indentation test results should be analysed, and the test results cannot be simply listed.

Response 4: The tensile test results have been analysed in great detail. However, I agree that a discussion of the micro-indentation test results was missing, I therefore, added it to the discussion at the end of the paragraph.

Reviewer 2 Report

In this paper by Hemes and colleagues, the authors studied the microstructure and mechanical properties of Ti6Al4V upon thermomechanical processing. The results fit the scope of Materials and they are a valuable contribution to the field of additive manufacturing. However, some issues must be tackled before the paper can be recommended for publication. Please see the comments below:
1) "In all cases, the hybrid samples consisted of a forged base plate with dimensions of 100 x 100 mm and a thickness of 10 mm, to which 50 x 50 x 100 mm³ of Ti6Al4V material was added by p-LMD as illustrated in Figure 2a" (Lines 73-75). - figures should be introduced in the text in the chronological order. There is no mention of Fig. 1. I suspect that the authors mean "illustrated in Figure 1a". Please make the necessary corrections.
2) Characterization details such as SEM acceleration voltage, XRD range, etc. should be reported to make the study reproducible.
3) Captions should be on the same page as the corresponding figure (Lines 138-142).
4) Scale bar markers in Fig. 4 are barely visible. Moreover, the order of panels is counterintuitive - how come there is empty space in the middle? It would be beneficial to put the descriptors of samples next to the corresponding SEM micrographs to facilitate the interpretation of data. With this amount of panels, it is challenging to go back and forth between the figures and the caption descriptions.
5) Headlines should not be separated from Tables (Line 247).
6) Comparing the length of sections Results vs Discussion, the latter is very short, so the article provided limited impact. Please deepen the analysis and extract more findings from the data.
7) Conclusions section should describe the impact of this work and provide a future outlook.
8) How many times the processing was conducted and the resulting samples were characterized? In other words, is the process reproducible? Did the authors try to test if different batches produced at the same parameters give consistent results?

Author Response

Answers to Reviewer no. 2:

In this paper by Hemes and colleagues, the authors studied the microstructure and mechanical properties of Ti6Al4V upon thermomechanical processing. The results fit the scope of Materials and they are a valuable contribution to the field of additive manufacturing. However, some issues must be tackled before the paper can be recommended for publication. Please see the comments below:

1) "In all cases, the hybrid samples consisted of a forged base plate with dimensions of 100 x 100 mm and a thickness of 10 mm, to which 50 x 50 x 100 mm³ of Ti6Al4V material was added by p-LMD as illustrated in Figure 2a" (Lines 73-75). - Figures should be introduced in the text in the chronological order. There is no mention of Fig. 1. I suspect that the authors mean "illustrated in Figure 1a". Please make the necessary corrections.

Response 1: The order of the figures was changed, according to the reviewer’s comment; actually, figure 2 was meant here, but the respective image was placed in Figure 1, now, to fit the order better, and appear at the right position in the text.

2) Characterization details such as SEM acceleration voltage, XRD range, etc. should be reported to make the study reproducible.

Response 2: Characterization details (SEM acceleration voltage, XRD range, working distance etc.) were added to the Methods section.

3) Captions should be on the same page as the corresponding figure (Lines 138-142).

Response 3: The order of the text and figures was changed accordingly.

4) Scale bar markers in Fig. 4 are barely visible. Moreover, the order of panels is counterintuitive - how come there is empty space in the middle? It would be beneficial to put the descriptors of samples next to the corresponding SEM micrographs to facilitate the interpretation of data. With this amount of panels, it is challenging to go back and forth between the figures and the caption descriptions.

Response 4: The scale bars in Figure 4 (actually, Figure 5, now) have been increased. Moreover, the order of the images was changed to be more intuitive. I hope that the images, together with captions, will be clear and understandable, now.

5) Headlines should not be separated from Tables (Line 247).

Response 5: The order of the text and Tables was changed accordingly.

6) Comparing the length of sections Results vs Discussion, the latter is very short, so that the article provided limited impact. Please deepen the analysis and extract more findings from the data.

Response 6: The discussion of the results obtained in this study was deepened significantly, to give sufficient discussion of the results and to increase the impact of the article.

7) Conclusions section should describe the impact of this work and provide a future outlook.

Response 7: The conclusions were stated more clearly, expecting to highlight the impact of this work more clearly. Moreover, a future outlook was added.

8) How many times the processing was conducted and the resulting samples were characterized? In other words, is the process reproducible? Did the authors try to test if different batches produced at the same parameters give consistent results?

Response 8: Unfortunately, the experiments in total have not been repeated so far. Further testing should therefore, be carried out in the future, to check the representability of the results obtained. I added this to the outlook given.

Reviewer 3 Report

The reviewer comments of the paper «Microstructures and mechanical properties of hybrid, additively manufactured Ti6Al4V after thermomechanical processing»

- Reviewer

The authors presented an article «Microstructures and mechanical properties of hybrid, additively manufactured Ti6Al4V after thermomechanical processing». However, there are several points in the article that require further explanation.

Comment 1:

Introduction.

It is useful to add a paragraph for industrial applications of additively manufactured Ti6Al4V. Show the advantages and disadvantages in comparison with the classical method of obtaining. How does the production time of a product differ by different methods? Is the AM product suitable for all standard sizes? What parameters of surface roughness can be obtained? How much does the hardness of the products increase? Add an article in the introduction: 10.3390/ma12071006

Comment 2:

  1. Materials and Methods

What equipment is used for LMD? Bring the installation photo. Show the main elements. For devices and machine used in research, indicate in parentheses (manufacturer, city, country).

Explain why the parameters indicated in table 1 were selected?

Show on figure the tensile specimen with dimensions.

Comment 3:

The quality and resolution of the figures 1, 2, 3, 4 needs to be improved.

Better to do general section 3. Results and discussion

In section 3.3.1 add color stress and strain curves for all 4 specimen materials.

Comment 4:

It will be useful to add a section of Nomenclature in which to sign all the physical quantities and abbreviations encountered in the article. There are many physical quantities in the text and such a section will help to find the description of the necessary element.

For example,

Rp0.2       : Stress resulting in a plastic strain of 0.2%

TMP      : Thermomechanical processing

etc.

Comment 5:

Conclusions.

It is necessary to more clearly show the novelty of the article and the advantages of the proposed method. What is the difference from previous work in this area? Show practical relevance. What is the difference from other researchers? What are the quantitative and qualitative research results obtained?

Conclusions should reflect the purpose of the article.

Use the format:

  • Conclusions 1
  • Conclusions 2
  • Etc.

The article is interesting and written at a good scientific level. Authors should carefully study the comments and make improvements to the article step by step. After major changes can an article be considered for publication in the "Materials".

Author Response

Answers to Reviewer no. 3:

Reviewer

The authors presented an article «Microstructures and mechanical properties of hybrid, additively manufactured Ti6Al4V after thermomechanical processing». However, there are several points in the article that require further explanation.

Comment 1:

Introduction

It is useful to add a paragraph for industrial applications of additively manufactured Ti6Al4V. Show the advantages and disadvantages in comparison with the classical method of obtaining. How does the production time of a product differ by different methods? Is the AM product suitable for all standard sizes? What parameters of surface roughness can be obtained? How much does the hardness of the products increase? Add an article in the introduction: 10.3390/ma12071006

Response 1: The introduction was extended, including also the suggested article and findings herein. Moreover, the questions addressed above (advantages vs. disadvantage, surface roughness, hardness, production time etc.) have been answered to a large extend within the introduction or they were addressed in the discussion/outlook sections of the article, respectively.

Comment 2:

Materials and Methods

What equipment is used for LMD? Bring the installation photo. Show the main elements. For devices and machine used in research, indicate in parentheses (manufacturer, city, country).

Explain why the parameters indicated in table 1 were selected?

Show on figure the tensile specimen with dimensions.

Response 2: A photograph of the p-LMD/L-DED installation was added, hoping to provide sufficient information. Moreover, the manufacturers of devices and machines were added. An explanation for the selection of process parameters was given as well. And an image of a tensile specimen, together with locations of origin, orientations and dimensions is provided in an additional figure.

Comment 3:

The quality and resolution of the figures 1, 2, 3, 4 needs to be improved.

Better to do general section 3. Results and discussion

In section 3.3.1 add color stress and strain curves for all 4 specimen materials.

Response 3: Results and discussion sections were joined into one general section. Moreover, the quality of Figures 1-5 was improved. Unfortunately, I do not have the stress-strain curves of the tensile tests conducted, since they were carried out with our industrial project partners, whom provided me with the test results.

Comment 4:

It will be useful to add a section of Nomenclature in which to sign all the physical quantities and abbreviations encountered in the article. There are many physical quantities in the text and such a section will help to find the description of the necessary element.

For example,

Rp0.2 : Stress resulting in a plastic strain of 0.2%

TMP : Thermomechanical processing

etc.

Response 4: A table, containing all the nomenclature of the study was added to the introduction section.

Comment 5:

Conclusions

It is necessary to more clearly show the novelty of the article and the advantages of the proposed method. What is the difference from previous work in this area? Show practical relevance. What is the difference from other researchers? What are the quantitative and qualitative research results obtained?

Conclusions should reflect the purpose of the article.

Use the format:

    Conclusions 1

    Conclusions 2

    Etc.

Response 5: Conclusions have been extended and hopefully, clarified, to show the novelty of the article and the advantages of the proposed method, as well as highlighting differences to previous work and the practical relevance of the findings obtained.

The article is interesting and written at a good scientific level. Authors should carefully study the comments and make improvements to the article step by step. After major changes can an article be considered for publication in the "Materials".

Round 2

Reviewer 2 Report

I am satisfied with the presented corrections. The paper can be accepted for publication. 

Reviewer 3 Report

The authors have improved the article according to the comments. The article can now be published.